# Metabolic Oscillations and Glycolytic Phenotypes of Cancer Cells

**DOI:** 10.3390/ijms241511914

**Published:** 2023-07-25

**Authors:** Takashi Amemiya, Kenichi Shibata, Tomohiko Yamaguchi

**Affiliations:** 1Graduate School of Environment and Information Sciences, Yokohama National University (YNU), 79-7 Tokiwadai, Hodogaya-ku, Yokohama 240-8501, Japan; shibata-kenichi-wp@ynu.ac.jp; 2Meiji Institute for Advanced Study of Mathematical Sciences (MIMS), 4-21-1 Nakano, Nakano-ku, Tokyo 164-8525, Japan; tomosan@meiji.ac.jp

**Keywords:** glycolytic oscillations, cancer cells, glycolytic phenotype, malignancy, mathematical model, feedback inhibition

## Abstract

Cancer cells show several metabolic phenotypes depending on the cancer types and the microenvironments in tumor tissues. The glycolytic phenotype is one of the hallmarks of cancer cells and is considered to be one of the crucial features of malignant cancers. Here, we show glycolytic oscillations in the concentrations of metabolites in the glycolytic pathway in two types of cancer cells, HeLa cervical cancer cells and DU145 prostate cancer cells, and in two types of cellular morphologies, spheroids and monolayers. Autofluorescence from nicotinamide adenine dinucleotide (NADH) in cells was used for monitoring the glycolytic oscillations at the single-cell level. The frequencies of NADH oscillations were different among the cellular types and morphologies, indicating that more glycolytic cancer cells tended to exhibit oscillations with higher frequencies than less glycolytic cells. A mathematical model for glycolytic oscillations in cancer cells reproduced the experimental results quantitatively, confirming that the higher frequencies of oscillations were due to the higher activities of glycolytic enzymes. Thus, glycolytic oscillations are expected as a medical indicator to evaluate the malignancy of cancer cells with glycolytic phenotypes.

## 1. Introduction

Cancer cells are metabolically reprogrammed and enhance pathways of nutrient acquisition and metabolism to sustain replication and metastasis during tumorigenesis and development [1,2]. This altered metabolism is one of the hallmarks of cancer [3]. The classical example of reprogrammed metabolism in cancer cells is to enhance the glycolytic pathway, known as the Warburg effect or aerobic glycolysis [4]. Warburg and coworkers in the 1920s observed that rat-liver carcinoma tissues had an approximately tenfold increase in glucose to lactate conversion as compared to normal tissues, even in the presence of oxygen. Warburg originally hypothesized that the enhancement of glycolysis was due to mitochondrial dysfunction in cancer cells [5]. However, it is now realized that cancer cells have active and functional mitochondria, contrary to Warburg’s hypothesis [6]. Nonetheless, aerobic glycolysis has been observed in a variety of cancer types, including liver cancers [7], lung [8], breast [9], and high-grade glial tumors [10]. In addition, studies of glycolysis-related gene analysis have revealed that several types of cancers, including kidney renal clear-cell carcinoma (KIRC), head and neck squamous-cell carcinoma (HNSC), and lung squamous-cell carcinoma (LUSC) defined the glycolytic cancer group [11].

Although aerobic glycolysis has been observed in many cancer cells, most cancer cells do not use aerobic glycolysis alone. A significant role of oxidative phosphorylation (OXPHOS) has been highlighted in cancer metabolism in the past two decades [12,13,14]. The hybrid enhancement of glycolysis and OXPHOS has also been addressed in cancer progression and metastasis [12,13,14]. Evidence shows that the mitochondrial pathways are also reprogrammed to meet the high energy demands and biomass synthesis in cancer cells [13]. Furthermore, not a few cancer cells such as cervical, breast, hepatoma, and pancreatic cancer cells can switch from aerobic glycolysis to OXPHOS under limiting glucose conditions [15]. 

During extensive studies on metabolism in cancer cells since the 1920s, Ibsen and Schiller, in 1967, found oscillations of nucleotides and glycolytic intermediates in aerobic suspensions of starved Ehrlich ascites tumor cells upon the addition of glucose [16]. Though glucose has been found to induce oscillations of reduced pyridine nucleotides and glycolytic intermediates in yeast cells [17] and heart muscle extracts [18], such oscillations—called metabolic oscillations or glycolytic oscillations—have not been reported in cancer cells. 

Since the first study by Ibsen and Schiller, no studies have been published on glycolytic oscillations in cancer cells until 2017 when we observed the oscillations in HeLa cervical cancer cells at the single-cell level by using a monolayer cell system [19]. One of the reasons for the few studies on glycolytic oscillations in cancer cells is probably due to the low degree of their synchronization [19]. No oscillations or only ambiguous oscillations can be observed if the degree of cellular synchronization is low under a conventional cell suspension system [16,20]. Contrarily, a monolayer cell system [21], which was first developed for yeast glycolytic oscillations, enabled us to observe the oscillations in individual cancer cells, even when they were not synchronized [19,22]. 

Here, we focus on the glycolytic phenotype in cancer cells and present their experimental and modeling results of glycolytic oscillations in two types of cancer cells: HeLa cervical cancer cells and DU145 prostate cancer cells. In the case of HeLa cells, different types of cellular morphologies were also investigated: spheroids [23] and monolayers. Spheroids are dense, 3D aggregates of cells that adhere to each other via cadherins [24]. We can expect different characteristics of glycolytic oscillations in spheroidal and monolayered cancer cells because gene expression, drug sensitivity, and metabolic properties are different between spheroidal and monolayered cancer cells [25,26,27]. 

To get insights into the experimental results, we present a new mathematical model for glycolytic oscillations in cancer cells based on the enzymatic feedback reactions in the glycolytic pathway [28,29], which quantitatively reproduced the oscillatory behaviors. In summary, biomedical implications could be derived from the experimental and modeling results of oscillatory dynamics in cancer cells in terms of their metabolic phenotypes.

## 2. Mathematical Model

We modified a previously proposed modeling scheme [23,29] to simulate oscillations in HeLa and DU145 cells in monolayers and HeLa cells in spheroids. The present model has five variables, which are the same as those in the previous model, and newly includes the inhibition of phosphofructokinase (PFK) by lactate (Y), as shown in Figure 1. PFK is one of the most important regulatory enzymes in glycolysis and is known to be activated by ADP and inhibited by both ATP and metabolites such as phosphoenolpyruvate (PEP) and lactate, which are produced in the downstream of glycolytic pathway [30,31,32]. 

Briefly, the previous model considered two main processes in the glycolytic pathway: the upstream (ATP-consuming) reactions of hexose and the downstream (ATP-producing) reactions of triose. The two reactions were modeled by PFK– and pyruvate kinase (PK)–allosteric reactions, respectively; PFK was activated by ADP and inhibited by ATP, and PK was inhibited by ATP. In addition, the present model includes inhibition of PFK by intracellular lactate [32] to simulate the experimentally observed oscillatory behaviors in which the oscillations stopped, even though the concentrations of extracellular glucose were high enough to sustain the oscillations, as shown below. 

Several glycolytic reactions are lumped into the present model; the variable X after the PFK reaction denotes the concentrations of pools of intermediates from fructose 1,6-bisphosphate (F1,6BP) to 1,3-bisphosphoglycerate (1,3BPG) in the glycolytic pathway and the variable Y denotes the concentration of lactate as shown in Figure 1. The other variables are as follows: G, glucose; $G_{ex}$ extracellular glucose; $A_{3}$, ATP; and $Y_{ex}$, extracellular lactate.

The rate laws of six variables (*G*, *X*, *Y*, $A_{3}$, $G_{ex}$, and $Y_{ex}$), three transport kinetics ($J_{in}$, $J_{GLUT}$, and $J_{P,Y}$), and four reaction rates ($v_{1}$, $v_{2}$, $v_{3}$, and $v_{4}$) in the model are summarized in Table 1.

The four of each rate constant ($k_{1}-k_{4}$) contain constant values ($a_{1}-a_{4}$), respectively, and a common parameter α (>0) that may take different values for different cells [23,33,34,35]: (1)$$k_{i}=\alpha a_{i},\;\;\;i=1,\;2,\;3,\;4,\;\;$$

Equation (1) assumes that the values of the four rate constants can be different among cancer cells; however, they are not completely random. Notably, a higher value of α yields a higher rate of each reaction step, indicating a higher glycolytic activity in a cell. Thus, Equation (1) excludes the case where the activity of a particular enzyme is very high (or low), whereas the activities of the other enzymes are very low (or high) in a cell. This indicates that the activities of some glycolytic enzymes are often regulated synchronously [35]. 

In addition, the activity of the glucose transporter was expressed by β as,
(2)$$\beta =V_{max}/0.65\;\;$$
where the maximum glucose uptake rate $V_{max}$ ($mMs^{-1}$) was normalized to the value of 0.65 $mMs^{-1}$ for HeLa cells [29]. 

Notably, the parameter values, such as the rate constants and maximum glucose uptake rate, were set at 25 °C in the previous model [29]. Thus, these values (α and β) could be approximately twice as high as those at 37 °C [23] if the effect of temperature on the rate constants was considered. This assumption is based on studies showing that the Arrhenius-type dependence of the rate constant on the temperature can model reaction rates in biological reactions including metabolic processes [36,37,38]. The Arrhenius-type model contributes 1.8–3.0 to the $Q_{10}$ values: the change in reaction rates with every 10 K change in temperature. Table 2 summarizes the parameter values used for the calculations.

Table 3 summarizes the initial concentrations of variables and the total concentration of ATP and ADP used for the calculations. The initial concentration (0.30 mM) of glucose was assigned referring to a reported value (0.12 mM) in starved HeLa cells [39] and the other initial values were the same as those in our previous modeling studies [23,29]. The total concentration (3.0 mM) of ATP and ADP was assigned referring to a value (4.0 mM) in yeast cells [28]. 

## 3. Results

### 3.1. Glycolytic Oscillations in Monolayers of HeLa Cervical and DU145 Prostate Cancer Cells

The glycolytic oscillations were first investigated using different cancer cells, HeLa cervical cancer cells and DU145 prostate cancer cells, cultured in monolayers. Figure 2 shows the time course of glycolytic oscillations observed through NADH fluorescence signals from HeLa and DU145 cells. These cells were cultured in the high-glucose (25 mM) medium with antibiotics and then starved of glucose before the initiation of oscillations by adding 20 mM glucose at 25 °C. When the cells were not starved of glucose, no oscillations were observed, as shown in Figure 2A, as an example of HeLa cells. On the other hand, when the cells were starved of glucose, they exhibited oscillations lasting up to 450 s. The amplitude of oscillations was smaller in HeLa cells (Figure 2B) than those in DU145 cells (Figure 2C).

Statistical analysis of the frequency of oscillations revealed that most HeLa cells oscillated at higher frequencies than DU145 cells, with frequencies ranging 0.019–0.070 Hz with a median of 0.034 Hz in HeLa cells (Figure 2D) and ranging 0.0050–0.065 Hz with a median of 0.023 Hz in DU145 cells (Figure 2E). 

To investigate the reason why the oscillations stopped, the amount of glucose was measured by using the urinary glucose test strip after stopping the oscillations. Since the test strip showed green in color, which indicated 28 mM glucose in this semiquantitative test (see Section 5), the glucose concentration after the fluorescent observation was high enough to exhibit oscillations; this was the same for both HeLa and DU145 cells and all the cases hereafter. This information was used in the numerical simulation of the oscillations using a mathematical model.

### 3.2. Comparison of Spheroids and Monolayers of HeLa Cells

To investigate the oscillatory behaviors in different cellular morphologies, we used HeLa cells in spheroids and monolayers. Figure 3 shows the time course of NADH fluorescence in HeLa cells in the spheroids and in monolayers, respectively. These cells were cultured in a low-glucose medium (5.6 mM) without antibiotics and starved with both glucose and FBS before the initiation of glycolytic oscillations by adding 25 mM glucose at 37 °C.

Figure 3A shows the oscillations in seven single cells in spheroids. The oscillations in HeLa cells in spheroids (Figure 3A) were more irregular and heterogeneous than those in monolayers (Figure 3B). Notably, HeLa cells in both spheroids and in monolayers exhibited oscillations even without the above starvation process under the present conditions, although the fraction of oscillating cells was very small (less than 0.01).

A statistical analysis of the frequency of oscillations revealed that most of the HeLa cells in spheroids oscillated with higher frequencies than in monolayers; their frequencies of oscillations ranged 0.031–0.17 Hz with a median of 0.070 Hz in spheroids (Figure 3C) and ranged 0.031–0.078 Hz with a median of 0.031 Hz in the monolayers (Figure 3D), respectively. The fraction of oscillating ROIs in a spheroid was considered to be the same as that of oscillating cells in the spheroid (see Materials and Methods) and was used to calculate the distribution of frequency in HeLa cells in spheroids (Figure 3C).

Notably, the glucose concentration after the fluorescent observation was high enough to exhibit oscillations after the oscillations were stopped for HeLa cells in both spheroids and monolayers by testing the urinary glucose test strips, as mentioned above. 

### 3.3. Numerical Simulations of Glycolytic Oscillations in HeLa and DU145 Cells

Our previous mathematical model included the PFK and PK reactions as key enzymatic reactions regulating the upstream and downstream of the glycolysis, respectively referring to a glycolytic model for yeast cells [40]. The previous model succeeded in reproducing the frequency of oscillations in HeLa cells, however, it could not reproduce the duration of oscillations [23,29]. Namely, when 20 mM or 25 mM glucose was used as the initial and added concentrations, the duration of the oscillations was much longer than that in the experiments. 

To investigate the mechanism of stopping the oscillations, the present model newly included feedback inhibition of PFK by lactate produced in the glycolysis and excreted to extracellular space through a monocarboxylate transporter (MCT). Thus, we expect that the oscillations can be stopped before most of the extracellular glucose is consumed by the cells. 

Thus, numerical simulations were performed using the present mathematical model, with the parameter values listed in Table 2. The activity of glycolytic enzymes α (Equation (1)) and glucose transporter β (Equation (2)) were used as the two parameters for the calculations. We can see from the phase diagram (Figure 4A) that higher activities of glycolytic enzymes yielded higher frequencies of the oscillations, although GLUT activity did not significantly affect the frequencies.

The numerically simulated results for glycolytic oscillations in HeLa and DU145 cells are shown in Figure 4B. The appropriate choice of parameters, α and β, as indicated in Figure 4A, agreed well with the frequencies of oscillations in HeLa and DU145 cells in the monolayers, as well as HeLa cells in spheroids and monolayers (Figure 4B). The experimental median frequencies, as shown in Figure 2 and Figure 3, and the simulated frequencies, as shown in Figure 4B, are summarized in Table 4. The simulated results indicated that the enhancement of glycolytic enzymes was higher in HeLa cells than in DU145 cells in the monolayers and was also higher in HeLa cells in spheroids than in monolayers (c and d).

### 3.4. Effect of the Inhibitory Feedback Mechanism on the Glycolytic Oscillations

To get more insights into the effect of the newly added inhibition of PFK by lactate (variable Y in Figure 1), we carried out numerical simulations with and without this inhibition mechanism for the experimental (Figure 2B) and modeling (panel a in Figure 4B) results of HeLa cells in the monolayers. As shown in Figure 5, the oscillatory durations were significantly longer (more than 2000 s) without the lactate inhibition (Figure 5B) than those in the experiments (350–400 s), as well as in the calculations with the lactate inhibition (approximately 380 s, Figure 5A). 

The calculated oscillations with a frequency of 0.035 Hz stopped at around 380 s when the added glucose was still present at a concentration of approximately 20 mM in the extracellular solutions (Figure 5A), which was consistent with the experimental observations of the urinary glucose test, as mentioned above. 

## 4. Discussion

The present study aimed to characterize glycolytic phenotypes of cancer cells from oscillatory behaviors in glycolysis, called glycolytic oscillations. The oscillations were demonstrated in two types of cancer cells, HeLa cervical cancer cells and DU145 prostate cancer cells, and in two types of cellular morphologies, spheroids and monolayers. The median frequency was higher in HeLa cells (0.034 Hz) than in DU145 cells (0.023 Hz), and was higher in HeLa spheroidal cells (0.070 Hz) than in the monolayered cells (0.031 Hz). A new mathematical model for glycolytic oscillations in cancer cells was presented, based on the enzymatic feedback reactions in the glycolytic pathway. The model reproduced the experimental oscillatory behaviors quantitatively, indicating that the higher frequency of oscillations was due to the higher rate of enzymatic reactions in the glycolysis. 

Metabolic oscillations have also been uncovered in a comprehensive model including both the cancer-related metabolic pathway and the gene regulatory network [2]. The timescale of oscillations in their model is much greater than the timescale in Ehrlich ascites tumor cells in cell suspensions [16] or in HeLa and DU145 cells in the monolayers and spheroids in this study. Notably, the oscillations in their model occur among the metabolically normal state, the cancer OXPHOS state, and the cancer glycolysis state [2]. They are caused by the long time-period regulations between genes and metabolites. On the other hand, the experimentally observed oscillations in the Ehrlich tumor [16] and the present cancer cells are due to the enzymatic kinetics in the glycolytic pathway and are not involved in the regulation of genes. 

In the experiments, glucose starvation prior to glucose addition could induce glycolytic oscillations in cancer cells. When starved, cancer cells activate AMP-activated protein kinase (AMPK) under glucose-depleted conditions [41,42] to maintain ATP levels; cancer cells switch from anabolic to catabolic metabolism by stimulating glucose uptake, aerobic glycolysis, and ATP synthesis [42]. In particular, glucose uptake was reported to increase by 23 times by glucose starvation in HeLa cells [43]. Such a large increase in glucose uptake could induce glycolytic oscillations in cancer cells, as shown in the phase diagram in Figure 4A. 

Metabolites can activate or inhibit the enzymes in their generation pathways. PFK, one of the most important regulatory enzymes of glycolysis, is allosterically activated by ADP and fructose 2,6-bisphosphate and inhibited by ATP, phosphoenolpyruvate, and lactate [30,31,32]. In addition, HK is inhibited by glucose 6-phosphate [44] and PK is activated by fructose 1,6-bisphosphate [45] and inhibited by ATP [46]; both HK and PK are also key regulatory enzymes of glycolysis. Among the important regulatory mechanisms in glycolysis, our previous model for glycolytic oscillations in cancer cells took into account ADP-activation and ATP-inhibition of PFK as well as ATP-inhibition of PK, referring to glycolytic oscillation models for yeast cells [28,40]. 

Though the previous mathematical model could reproduce the frequency of oscillations quantitatively, it could not reproduce the duration of oscillations; the model yielded a longer duration of oscillations than that in the experiments when external glucose of 25 mM was used in the model (Figure 5B). Meanwhile, the present model has a newly added inhibition of PFK by intracellular lactate (variable Y in Figure 1) to the previous model and could reproduce both the frequency and duration of oscillations in the experiments quantitatively (Figure 5A). The present mathematical model also demonstrates that the cause of stopping oscillations is not the consumption of all extracellular glucose (Figure 5A and the experimental urinary glucose test) but the inhibition of PFK by intracellular lactate produced in glycolysis. 

The calculated concentrations of intracellular lactate were approximately 1.5–3 mM (Y in Figure 5A) for monolayered HeLa cells (Figure 2B and panel a in Figure 4B) and approximately 2.0–3.5 mM (not shown) for spheroidal HeLa cells (Figure 3A and panel c in Figure 4B); the lactate concentrations were a little larger in more glycolytic-spheroidal HeLa cells than monolayered HeLa cells. Real values of lactate concentrations are reported to range between approximately 1–2 mM in normal/immortalized cells and 4–9 mM in tumor-derived cells [47], as well as between 4–12 mM in highly metastatic MDA-MB−231 breast cancer cells and 1.4–3.0 mM in lowly metastatic MCF7 breast cancer cells in the case of low glucose/glutamine conditions [48]. When provided with a medium having 25 mM glucose/4 mM glutamine, both MDA-MB−231 and MCF7 cells showed high intracellular lactate levels of 11 mM [48]. Though the conditions are different between our study and the reported ones, the calculated lactate concentrations can be said to represent the real values of lactate concentrations in cancer cells. 

Different oscillation frequencies were observed depending on the cell type (Figure 2) and the cellular morphology in HeLa cells (Figure 3). It has been reported that the activities of all glycolytic enzymes are enhanced in cancer spheroids; spheroidal HeLa cells undergo significantly enhanced glycolysis compared with those in monolayers [25,26] and the enhanced glycolysis was also observed in other tumor spheroids [25,26,49]. This was likely the main reason for the higher frequency of oscillations in HeLa cells in spheroids than in the monolayers in the present study. In addition, ovarian cancer spheroid cells have been reported to be more aggressive in growth, migration, and invasion, and more resistant to chemotherapy [49]. Thus, the frequency of glycolytic oscillations can be a good indicator for evaluating the malignancy of cancer cells. 

The present modeling study revealed that the higher frequency of oscillations was due to higher glycolytic enzymatic activities in cancer cells (Figure 4). Enhanced glycolytic activity is a crucial factor in the malignant phenotype of cancers [50]. For instance, the triple-negative breast cancer (TNBC) cells—which are malignant and defined as a subtype of breast cancer cells that are the absence of estrogen receptors (ER), progesterone receptors (PR), and human epidermal growth factor receptor−2 (HER2)—exhibited a highly glycolytic phenotype with high glucose uptake, increased lactate production, and low mitochondrial respiration compared with ER-positive cells [51]. In addition, studies of nonsmall cell lung cancer (NSCLC) which consists of approximately 85%–88% of lung cancers showed that patients with NSCLC exhibiting higher values of whole-body total lesion glycolysis (TLG) demonstrated lower overall survival rates than those with NSCLC exhibiting lower TLG values [52]. Furthermore, comprehensive metabolic profiling of pancreatic ductal adenocarcinoma (PDAC) revealed that the glycolytic subtype is strongly associated with the mesenchymal subtype, which is typically more aggressive than the epithelial subtype and typically has an overall poor prognosis [53,54]. Collectively, the present mathematical model can be applied to evaluate the malignancy of cancer cells in terms of their glycolytic phenotypes. 

Mitochondrial membrane potential also oscillates through glucose or energy metabolism [55]. Little is known about mitochondrial oscillations and their interactions with glycolytic oscillations in cancer cells [56]. The present study assumes that intracellular glucose is metabolized to lactate without using mitochondrial respiration. This is based on the fact that the activity of the mitochondrial pyruvate carrier (MPC) is reduced in cancer cells [57]; thus, they mostly use glycolysis for ATP production when glucose is the only nutrient supply. 

Based on the above, as well as the previous studies of yeast glycolytic oscillations [17,21], this study also assumes that the NADH fluorescence results only from glycolysis. However, this should be inspected because NADH and NAD^+^ exist in different subcellular compartments, including cytosol and mitochondria. Thus, to measure the NADH levels in cytosol and to assess glycolytic rates actually, we should use, for instance, a fluorescence resonance energy transfer (FRET) biosensor which can separately monitor glycolytic and mitochondrial NADH levels, as well as the NAD^+^/NADH ratio [58]. 

Metabolic interactions may occur between cells. For instance, metabolic symbiosis has been proposed between glycolytic and oxidative cancer cells [59], as well as between cancer cells and cancer-associated fibroblasts (CAFs) [60]. In metabolic symbiosis, lactate is produced in glycolytic cancer cells or CAFs and then transported to oxidative cancer cells that metabolize it using OXPHOS; the latter cancer metabolism is called the reverse Warburg effect [61]. By metabolic symbiosis, cancer cells can acquire heterogeneities in energy metabolism in the tumor environment and reinforce their malignancy [60]. Studies of metabolic oscillations will also be able to reveal the mechanisms of metabolic symbiosis, which is a hot topic in cancer metabolism [62]. 

## 5. Materials and Methods

### 5.1. Cultures and Starvation of Glucose for HeLa Cervical and DU145 Prostate Cancer Cells in Monolayers

HeLa cervical cancer cells and DU145 prostate cancer cells were obtained from Cell Lines Service GmbH (Eppelheim, Germany) and the American Type Culture Collection (ATCC), respectively. Cells of each type were routinely cultured in Dulbecco’s modified Eagle’s medium (DMEM, FUJIFILM Wako Chemical Co., Osaka, Japan) and fetal bovine serum (FBS, 10% *v/v*; Hyclone, Cytiva, Tokyo, Japan) as described elsewhere [19,22]. Notably, the culturing medium contained high glucose (25 mM) and 1% antibiotic-antimycotic solutions. 

The procedures of fluorescence measurement were also described in the literature [19,22]. Briefly, monolayers of each type of cell were detached and then seeded in a well of a slide and chamber (Fukae-Kasei Co., Ltd., Watson Bio Lab, Kobe, Japan) at cell densities of $1.0\times 10^{5}$ to $2.0\times 10^{5}$ cells in 1 mL of the medium; the densities on the growth surface were $6.0\times 10^{4}$ to $1.2\times 10^{5}$ $cells/{cm}^{2}$. The seeded cells were incubated at 37 °C and 5% CO_2_ for 3 d and then starved of glucose at 37 °C and 5% CO_2_ for 24 h. The fluorescence microscopy was carried out in Dulbecco’s phosphate-buffered saline (DPBS; Sigma-Aldrich Co., LLC., Tokyo, Japan) at a pH of 6.9 at 25 °C (air-conditioned room temperature). DMEM was not used for the fluorescence measurement to avoid possible fluorescence from the amino acids and vitamins in it. The pH of 6.9 was selected from preliminary experiments in which the fraction of oscillating HeLa cells was relatively high. 

### 5.2. Cultures and Starvation of Glucose for HeLa Cervical Cancer Cells in Spheroids and Monolayers

HeLa cervical cancer cells obtained from the RIKEN BRC Cell Bank (RCB007) were used in this study. The cells were routinely cultured in the same way as mentioned above, except for DMEM with low glucose (5.6 mM) and the antibiotic antimycotic solutions. Antibiotics were not used in the present case because they inhibited spheroid formation, as reported in the literature [63]. Relier et al. inhibited spheroid formation by antibiotics in some cell lines [63]; however, the precise mechanism is not clear yet. The oscillatory behaviors in HeLa cells were compared between spheroids and in monolayers under the same experimental conditions. 

Procedures for the formation of HeLa cell spheroids are described in detail in our previous study [23]. Briefly, the cells detached and suspended in the culture medium were seeded in a well of ultralow attachment 96-well plates (PrimeSurface 96U; Sumitomo Bakelite Co., Ltd., Tokyo, Japan) and cultured at 37 °C and 5% CO_2_ for 30–48 h.

The starvation of spheroids was carried out under glucose-free conditions for 0 h to 52 h. Then, for fluorescence microscopy, the spheroids were put into a slide and chamber equipped with a MAS-GP Type A-coat slide and fixed by coating with alginate gel. 

In the case of cells cultured in monolayers, they were detached and cultured on a slide and chamber at 37 °C and 5% CO_2_ for 2 d. Then, the cells were incubated at 37 °C and 5% CO_2_ for 0 to 48 h in 100% glucose-free DMEM. Following glucose starvation, the medium in the wells was replaced with DPBS (pH 6.90 for fluorescence microscopy analysis).

### 5.3. Fluorescence Microscopy

An inverted fluorescence microscope (TC 5000; Meiji Techno Co. Ltd., Miyoshi, Japan) equipped with a thermoplate (TPi-CKX, Tokai Hit, Fujinomiya, Japan) was used for the fluorescence microscopy. NADH fluorescence was achieved by excitation with a 365 nm mercury line and emission was measured at 435–485 nm with a filter set (49000-ET-DAPI; Chroma Technology Corp., Bellows Falls, VT, USA). In the case of the first experiments using HeLa and DU145 cells in monolayers, fluorescence microscopy was performed at 25 °C (air-conditioned room temperature). The second experiments for HeLa cells in spheroids and monolayers were carried out at 37 °C by using the thermoplate.

Glycolysis was induced by the addition of glucose with a final concentration of 20 or 25 mM dissolved in DPBS at pH 6.90 and then monitored through the autofluorescence of intracellular NADH [23]. Procedures for image acquisition and data analysis are described in detail elsewhere [23]. Briefly, ImageJ [64] was used to analyze the fluorescence signals. To investigate the oscillatory behaviors of cells in the spheroids, we analyzed the NADH signals from a region of interest (ROI) which consisted of 5 × 5 pixels over the entire area of a spheroid. MATLAB (Math Works) was used for data processing and analysis unless otherwise stated [65,66].

Following fluorescence microscopy, residual glucose was checked with the urinary glucose test strip, New Uri-Ace Ga (Terumo, Corporation, Tokyo, Japan). Since its color chart has color matches at 0, 2.8, 5.6, 28, and 111 mM of glucose, the test indicates 28 mM if glucose consumption is low in the present study. 

The fluorescence data were obtained in three fields of view in two separate experiments for HeLa cells and DU145 cells in monolayers and in three fields of view in more than three separate experiments for HeLa cells in spheroids and monolayers. 

## Figures and Tables

**Figure 1 ijms-24-11914-f001:**
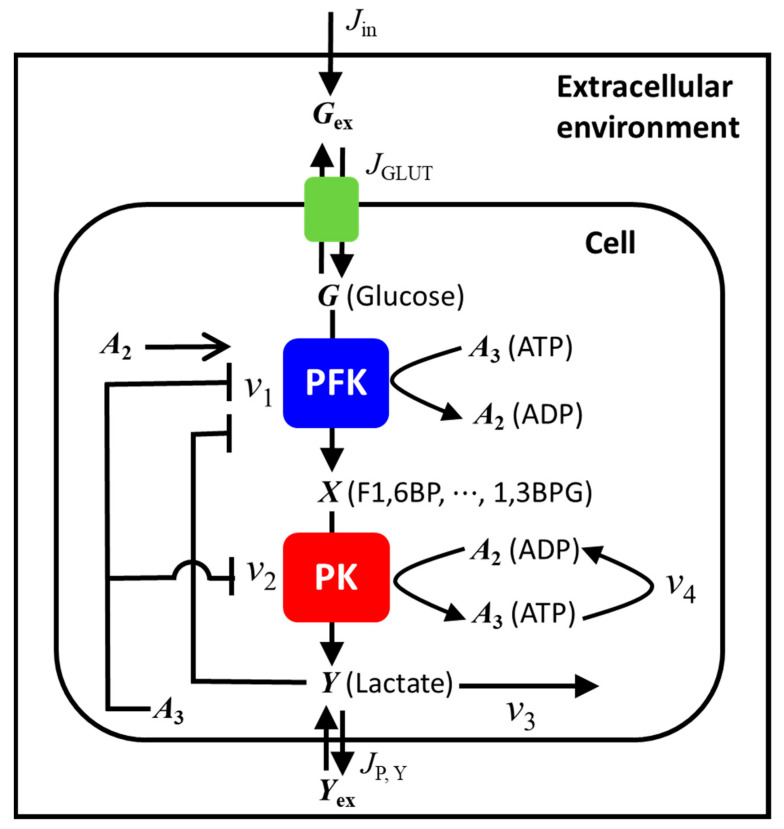
A mathematical model for the glycolytic oscillations in cancer cells. Variables: G, intracellular glucose; X, pool of intermediates following the PFK reaction; Y, lactate; $A_{3},$ ATP; $G_{ex}$, extracellular glucose; $Y_{ex}$, extracellular Y; $J_{in}$, glucose flux into the extracellular solution; $J_{GLUT}$, glucose transport through glucose transporter GLUT; $J_{P,Y}$, flux of difference in Y and $Y_{ex}$ through monocarboxylic transporter; $v_{1}$, reaction rate of PFK; $v_{2}$, reaction rate of PK; $v_{3}$, reaction rate of consumption of Y; $v_{4}$, reaction rate of consumption of ATP. PFK is allosterically activated by $A_{2}$ and inhibited by $A_{3}$ and Y. PK is allosterically inhibited by $A_{3}$. Abbreviations: PFK, phosphofructokinase; PK, pyruvate kinase; F1,6BP, fructose 1,6-bisphosphate; 1,3BPG, 1,3-bisphosphoglycerate.

**Figure 2 ijms-24-11914-f002:**
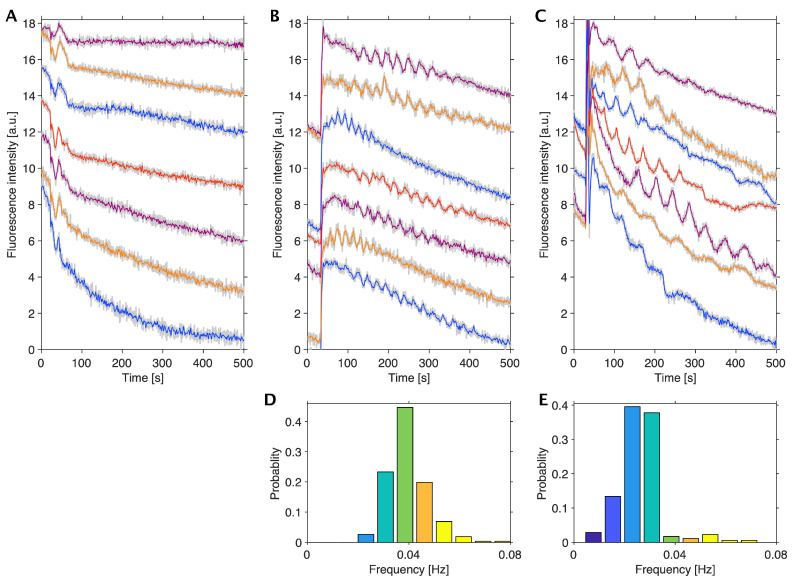
Glycolytic oscillations in HeLa and DU145 cells in monolayers. A time series of NADH fluorescence signals from single cells (**A**–**C**) and their frequency distributions (**D**,**E**): HeLa cells with no starvation of glucose (**A**), HeLa cells starved of glucose for 24 h (**B**), DU145 cells starved of glucose for 24 h (**C**), HeLa cells (N = 262 oscillatory cells/544 total cells) exhibiting median frequency of 0.034 Hz (**D**), and DU145 cells (N = 172 oscillatory cells/651 total cells) exhibiting median frequency of 0.023 Hz (**E**). Glucose of 20 mM was added to the cells at 30 s following each starving condition at 25 °C. Multiple oscillation curves in (**A**–**C**) are the typical examples of NADH fluorescence signals; different colors in the curves are ease of visibility to readers, the grey curves are original data, and the colored curves are their average. The color columns in panels (**D**,**E**) indicate the difference in the frequency ranges of oscillations.

**Figure 3 ijms-24-11914-f003:**
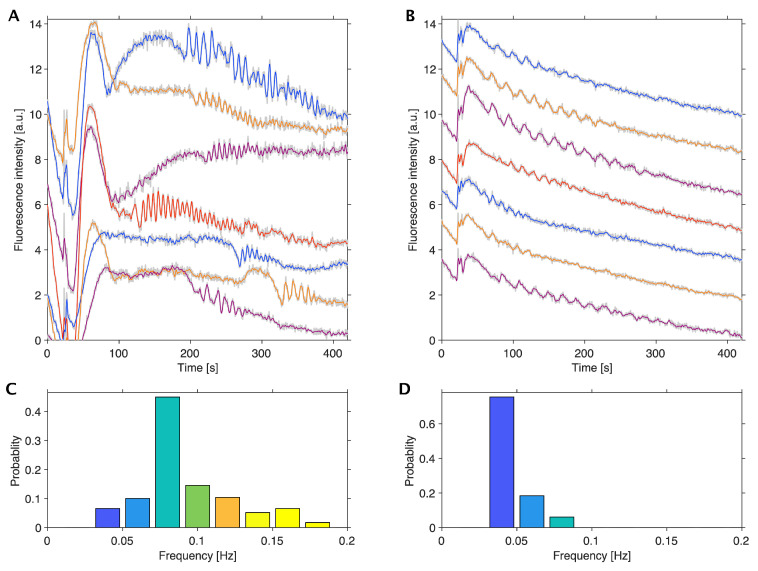
Comparison of glycolytic oscillations in spheroids and monolayers of HeLa cells ^a^. A time series of NADH fluorescence signals from single cells in spheroids (**A**), and in monolayers (**B**); and frequency distribution of oscillations in HeLa cells in spheroids (**C**) and in monolayers (**D**). Glucose (25 mM) was added to the cells at 30 s following starvation of both glucose and FBS for 24 h in spheroids (**A**) and for 2 h in monolayers (**B**) at 37 °C. The frequency distributions were calculated by ROIs (N = 289 oscillatory ROIs/563,162 total ROIs) from 74 spheroids starved of glucose and FBS for 0–52 h, exhibiting median frequency of 0.070 Hz (**B**), and by HeLa cells (N = 49 oscillatory cells/4948 total cells) from 22 monolayers starved of glucose and FBS for 0–2 h, exhibiting median frequency of 0.031 Hz (**D**), respectively. Multiple oscillation curves in (**A**,**B**) are the typical examples of NADH fluorescence signals; different colors in the curves are ease of visibility to readers, the grey curves are original data, and the colored curves are their average. The color columns in panels (**C**,**D**) indicate the difference in the frequency ranges of oscillations. (^a^ Modified from [23] according to the permission procedure of the Wiley journal Content).

**Figure 4 ijms-24-11914-f004:**
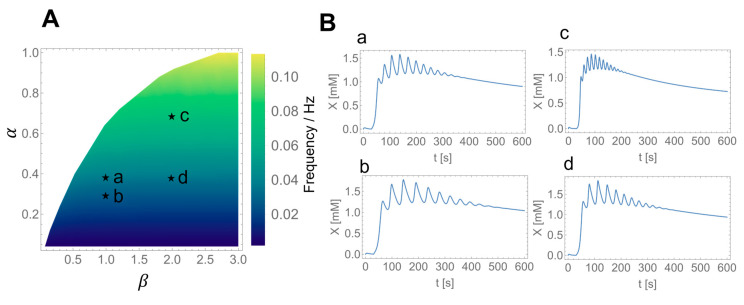
Numerical calculations of the mathematical model. (**A**) Phase diagram spanned by the enzymatic activity α (Equation (1)) and the GLUT activity β (Equation (2)). White area and blue-green-yellow gradation area indicate nonoscillatory and oscillatory states, respectively. The marks ⋆a−⋆d indicate points used for numerical simulation. (**B**) Time series of simulated oscillations in X for HeLa cells exhibiting frequency of 0.035 Hz (panel **a**), and DU145 cells exhibiting frequency of 0.024 Hz (panel **b**) in monolayers at 25 °C (Figure 2) calculated with α=0.37,β=1.0 (⋆a in panel **A**), and α=0.27,β=1.0 (⋆b in panel **A**), respectively; and for HeLa cells in spheroids exhibiting frequency of 0.072 Hz (panel **c**), and in monolayers exhibiting frequency of 0.034 Hz (panel **d**) at 37 °C (Figure 3) calculated with α=0.68,β=2.0 (⋆c in panel **A**), and α=0.37,β=2.0 (⋆d in panel **A**), respectively. The other parameter values are listed in Table 2.

**Figure 5 ijms-24-11914-f005:**
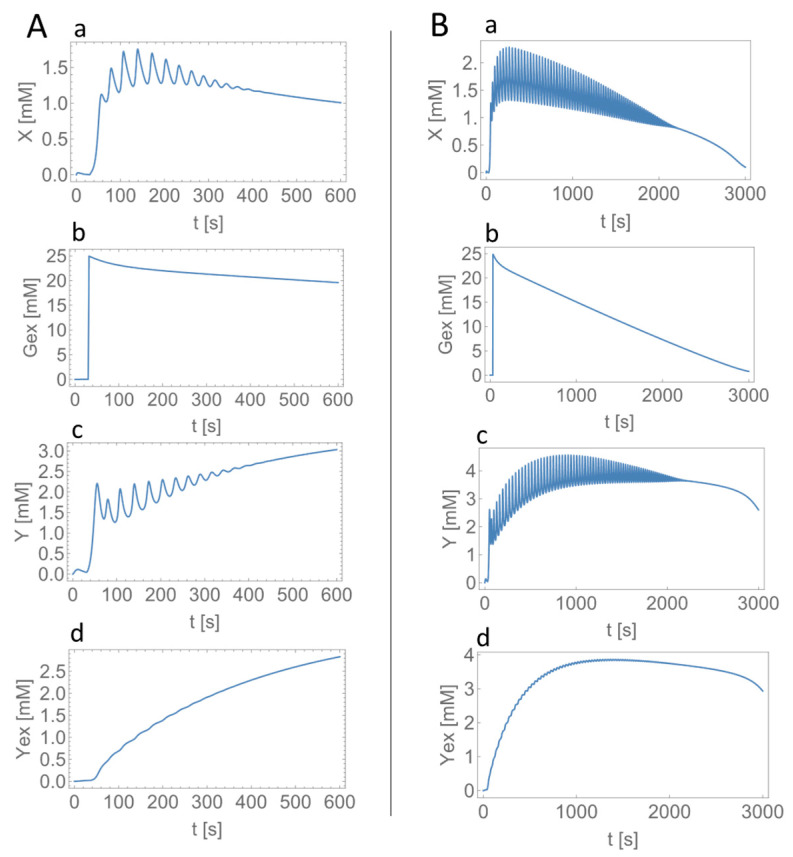
Effect of the feedback inhibition of PFK by lactate (Y) on the oscillatory behaviors in the model. Calculated time series in (**a**) X, (**b**) $G_{ex}$, (**c**) Y, and (**d**) $Y_{ex}$ with the inhibition of PFK by Y with α=0.37,β=1.0 (**A**) and without the inhibition of PFK by Y with α=0.49,β=1.0 (**B**). The frequencies in X are 0.035 Hz (**A**), and 0.036 Hz (**B**). Glucose of 25 mM was added at 30 s.

**Table 1 ijms-24-11914-t001:** Rate laws, transport kinetics, and reaction rates for the model.

Rate laws
$\frac{dG}{dt}=J_{GLUT}-v_{1}$ $\frac{dX}{dt}=v_{1}-v_{2}$ $\frac{dY}{dt}=2v_{2}-v_{3}-J_{P,Y}$ $\frac{dA_{3}}{dt}=-2v_{1}+4v_{2}-v_{4}$ $\frac{dG_{ex}}{dt}=J_{in}-\varphi J_{GLUT}$ $\frac{dY_{\mathrm{ex}}}{dt}=\varphi J_{P,Y}$
Transport kinetics
$J_{in}=\left\{\begin{array}{c} \frac{G_{in}}{t_{2}-t_{1}},\;\;\;\;\;\;\;t_{1}\leq t\leq t_{2}\\ 0,\;\;\;\;\;\;\;\;\;\;\;\;\;\;\;\;\;\;\;other \end{array}\right.$ $J_{GLUT}=\beta \frac{G_{ex}-\frac{G}{K_{eq}}}{K_{out}\left(1+\frac{G}{K_{in}}\right)+G_{ex}}$ $J_{P,Y}=\kappa (Y-Y_{ex})$
Reaction rates
$v_{1}={\alpha k}_{1}GA_{3}f\left(G,\;A_{3},\;Y\right),\;$ $\;\;with\;f\left(G,\;A_{3},\;Y\right)=\frac{{\left(A_{0}-A_{3}\right)}^{m}}{\left[1+{\left(A_{0}-A_{3}\right)}^{m}\left(\frac{1}{K_{1}}+\frac{G}{K_{1}K_{3}}+\frac{A_{3}}{K_{1}K_{4}}\right)+\frac{A_{3}^{m}}{K_{2}}+\frac{Y^{m}}{K_{8}}\right]}$ $v_{2}={\alpha k}_{2}X\left(A_{0}-A_{3}\right)g(X,\;A_{3})$, $\;\;with\;g\left(X,\;A_{3}\right)=\frac{1}{\left[1+\frac{A_{3}^{n}}{K_{5}}+\frac{X}{K_{6}}+\frac{\left(A_{0}-A_{3}\right)}{K_{7}}\right]}$ $v_{3}={\alpha k}_{3}Y$ $v_{4}={\alpha k}_{4}A_{3}$

Variables: G, intracellular glucose; X, pool of intermediates following the PFK reaction; Y, lactate; $A_{3},$ ATP; $G_{ex}$, extracellular glucose; $Y_{ex}$, extracellular Y; $J_{in}$, glucose flux into the extracellular solution; $J_{GLUT}$, glucose transport through glucose transporter GLUT; $J_{P,Y}$, flux of difference in Y and $Y_{ex}$ through monocarboxylic transporter. The stoichiometric coefficient 2 of $v_{2}$ in the dY/dt equation means that two molecules of Y (lactate) are produced from one molecule of glucose through the glycolysis. The stoichiometric coefficients −2 and 4 in the $dA_{3}/dt$ equation mean that two molecules of ATP are consumed, and four molecules of ATP are produced in the upstream and downstream of the glycolysis, respectively. The meaning and values of parameters are summarized in Table 2.

**Table 2 ijms-24-11914-t002:** The parameter values used for the calculations.

Model-Step	Parameters	Meaning	Values	Sources
$J_{in}$	$G_{in}$	External glucose additionfor 30 s≤t ≤32 s	20 or 25 mM	Experiments
$J_{GLUT}$	β	Normalized maximum glucose uptake	1.0 or 2.0	Equation (2)
	φ	Ratio of cellular volume to extracellular volume	0.1	[23,29]
	$K_{in}$	Michaelis constant of GLUT for intracellular glucose	12 mM	[23,29]
	$K_{out}$	Michaelis constant of GLUT for extracellular glucose	10 mM	[23,29]
	$K_{eq}$	Equilibrium constant	1.0	[23,29]
$J_{P,Y}$	κ	Transport constant of MCT	$0.1\;s^{-1}$	[23,29]
$v_{1}$	$k_{1}$	Rate constant of PFK reaction	$\alpha \;a_{1}$	Equation (1)
	α	A common parameter for the four rate constants	0.27–0.68	This work
	$a_{1}$	$A\;\mathrm{constant}\;\mathrm{value}\;\mathrm{for}\;k_{1}$	$1.0\;{mM}^{-(m+1)}\cdot s^{-1}$	[23,29]
	m	The number of substrate molecules bound to PFK	4	[23,29]
	$K_{1}$	Dissociation constant for free PFK and *m*-molecules of ADP	$1.0\;{mM}^{m}$	[23,29]
	$K_{2}$	Dissociation constant for free PFK and *m*-molecules of ATP	$1.0\;{mM}^{m}$	[23,29]
	$K_{3}$	Dissociation constant for ADP-activated PFK and glucose	1.0 mM	[23,29]
	$K_{4}$	Dissociation constant for ADP-activated PFK and ATP	1.0 mM	[23,29]
	$K_{8}$	Dissociation constant for free PFK and m-molecule of Y (lactate)	$2.0\;{mM}^{m}$	This work
$v_{2}$	$k_{2}$	Rate constant of PK reaction	$\alpha \;a_{2}$	Equation (1)
	α	A common parameter for the four rate constants	0.27–0.68	This work
	$a_{2}$	$A\;\mathrm{constant}\;\mathrm{value}\;\mathrm{for}\;k_{2}$	$0.5\;{mM}^{-1}\cdot s^{-1}$	[23,29]
	n	The number of substrate molecules Boud to PK	4	[23,29]
	$K_{5}$	Dissociation constant for free PK and *n*-molecule of ATP	$20\;{mM}^{n}$	[23,29]
	$K_{6}$	Dissociation constant for free PK and X (pool of intermediates)	20 mM	[23,29]
	$K_{7}$	Dissociation constant for free PK and ADP	20 mM	[23,29]
$v_{3}$	$k_{3}$	Rate constant of consumption of Y (lactate)	$\alpha \;a_{3}$	Equation (1)
	α	A common parameter for the four rate constants	0.27–0.68	This work
	$a_{3}$	$A\;\mathrm{constant}\;\mathrm{value}\;\mathrm{for}\;k_{3}$	$0.09\;s^{-1}$	[23,29]
$v_{4}$	$k_{4}$	Rate constant of consumption of ATP	$\alpha \;a_{4}$	Equation (1)
	α	A common parameter for the four rate constants	0.27–0.68	This work
	$a_{4}$	$A\;\mathrm{constant}\;\mathrm{value}\;\mathrm{for}\;k_{4}$	$0.15\;s^{-1}$	[23,29]

**Table 3 ijms-24-11914-t003:** Initial concentrations and total concentration ($A_{0}$) of ATP and ADP for the calculations.

Initial Concentrations		Sources
Variables	Values	
$G_{0}$	0.30 mM	[39]
$X_{0}$	0.30 mM	[23,29]
$Y_{0}$	0.30 mM	[23,29]
$G_{ex,\;0}$	0 mM	[23,29]
$Y_{ex,\;0}$	0 mM	[23,29]
Total concentration of ATP and ADP		
Constant	Value	
$A_{0}$	3.0 mM	[28]

**Table 4 ijms-24-11914-t004:** Summary of the experimental median frequencies (Exp.), as shown in Figure 2 and Figure 3, and the simulated frequencies (Sim.), as shown in Figure 4B.

Cells in Monolayers (at 25 °C)				HeLa Cells in Different Morphology (at 37 °C)			
HeLa Cells		DU145 Cells		Spheroids		Monolayers	
Exp. (Hz)	Sim. (Hz)	Exp. (Hz)	Sim. (Hz)	Exp. (Hz)	Sim. (Hz)	Exp. (Hz)	Sim. (Hz)
0.034	0.035	0.023	0.024	0.070	0.072	0.031	0.034

## Data Availability

The data that support the findings of this study are available from the corresponding author (amemiya-takashi-jk@ynu.ac.jp) upon reasonable request.

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
