# Peer review of "Metabolic Oscillations and Glycolytic Phenotypes of Cancer Cells"

_ijms, 2023, doi:10.3390/ijms241511914_

Round 1

Reviewer 1 Report

The manuscript, "Glycolytic Oscillations and Warburg Phenotypes of Cancer Cells" by Amemiya et al provides an analysis of NADH oscillations in HeLa and DU145 cells in adherent and spherical cultures following glucose stimulation. The authors follow their experimental results with a mathematical model of the system.

This study purports to demonstrate enhanced Warburg effect-like glycolysis in more glycolytic cancer cells compared to less glycolytic cells as implicated by higher NADH oscillating frequencies. The mathematical model further suggests that the results are due to differential activities of glycolytic enzymes. 

Overall, the work in this study is interesting. However, the following items must be addressed prior to publication:

1) The authors demonstrate a poor understanding of contemporary models of the "Warburg effect" and cancer metabolism. The citations related to the Warburg effect are out of date and do not reflect the current understanding of the field. The authors need to update their introduction (and the title of the paper) to reflect the fact that the field no longer believes that the Warburg effect is strictly operational in cancer cells. Indeed, even the authors cite more contemporary reviews that demonstrate that cancer cells consume oxygen in addition to engaging in high rates of glycolysis. Citing and summarizing the 2016 review by Chandel and DeBerardinis would be a good start.

2) The authors should cite other authors who have recently contributed to assessing various levels of glycolytic metabolism in cancer cells and different types of cancers. Citing and summarizing the following would be a good start: 

Feng, J.; Li, J.; Wu, L.; Yu, Q.; Ji, J.; Wu, J.; Dai, W.; Guo, C. Emerging Roles and the Regulation of Aerobic Glycolysis in Hepatocellular Carcinoma. J. Exp. Clin. Cancer Res. 202039, 126.

Vlassenko, A.G.; McConathy, J.; Couture, L.E.; Su, Y.; Massoumzadeh, P.; Leeds, H.S.; Chicoine, M.R.; Tran, D.D.; Huang, J.; Dahiya, S.; et al. Aerobic Glycolysis as a Marker of Tumor Aggressiveness: Preliminary Data in High Grade Human Brain Tumors. Dis. Markers 20152015, 874904.

Chacon-Barahona, J.A.; MacKeigan, J.P.; Lanning, N.J. Unique Metabolic Contexts Sensitize Cancer Cells and Discriminate between Glycolytic Tumor Types. Cancers 202315, 1158. https://doi.org/10.3390/cancers15041158

3) In the places where the authors report concentrations as g/L or g/mL, the authors should report concentrations as molarity (mM) in order to remain consistent throughout the manuscript.

4) In the results, the authors should moderate their language related to their conclusions. The authors have used a indirect measurement of NADH levels with no actual control to demonstrate that they are only (or actually) measuring NADH; yet they make claims that they are definitively assessing glycolytic rates. This is a stretch at best. The authors need to include a discussion on the legitimate and significant limitations of their study with respect to how confident they can be that they are actually measuring glycolytic rates. 

In places, English usage is subpar and unacceptable for publication.

Reviewer 2 Report

Comments to the manuscript

Summary

Glycolysis plays a prominent role in cancer metabolism and growth. Its dynamics have oscillatory activity, which can be monitored by autofluorescence of the cellular NADH. In this study, the frequencies of such oscillations were integrated into a mathematical model to simulate the activities of two regulatory glycolytic enzymes, phosphofructokinase (PFK) and pyruvate kinase (PK), of live tumor cells in conditions of glucose and serum replenishment following their respective deprivations. These data may be indicators of cancer cell heterogeneity and malignancy and thus provide knowledge fundamental for diagnosis and treatment of the cancer.

The combination of cellular fluorescence data and mathematical simulations provides a new level of information for metabolic modelling and is complementary to the numerous biochemical and genetic studies on cancer cells metabolism. This work should thus be of interest to both mathematicians in biomedical modelling, tumor cells biologists, biochemists, and preclinical scientists.

This manuscript is essentially well-written and structured. But there are several weaknesses which hamper the comprehension of the study.

General comments

Basically, this study provides novel results in comparing NADH oscillations in two different cell lines, assumed to have different glycolytic activities, and evaluated by applying a mathematical model which was modified to include a negative regulatory feedback of glycolytic metabolites on the activities of PFK and pyruvate kinase. The following points need to be considered:

1)     The title is a scientific over-interpretation, as the Warburg effect is not characterized by adequate (biochemical) terms for the cell models. It should, therefore, be modified (without the "Warburg effect").

2)     The text is substantially descriptive, but the paragraphs often lack an introductory thought (the aim of the experiment) and a conclusion or message from the described result. In general, the mathematical model needs more translation into biochemical terms of the glycolytic enzyme controls. The biochemistry should match the precision of the mathematical model. As the "Warburg effect" is biochemically ill-defined, please state to which glycolytic reaction/enzyme/metabolite you are referring.

3)     The manuscript is rather lengthy for the central message to be conveyed. It contains several aspects which are not necessary, redundant, or even distract from the line of thought (for example, extensive reference to OXPHOS, glycolytic oscillations in yeast, see below), and which are better contained in a review article and /or the Discussion. For minor points, one reference would suffice.

4)     A major issue is section 3.2., which is essentially previously published work, with the figure being identical (except for E,F). And it is not clear what is new about it in this context. Most parts of the Figure can be removed.

Specific comments to

1.                    Introduction

It is good style to refer previous publications  (i.e. on the Ehrlich ascites tumor cells) before highlighting one's own work. The emphasis being on glycolysis, referring to OXPHOS does not add necessary information for the focus of this work. Likewise, the section on yeast can be condensed; it would be more appropriate in the Discussion.

2.                    Mathematical Model

This section is poorly described. The definitions of the parameters and equations need improvement: For each mathematical parameter the corresponding biochemical parameter should be given. A table aligning the terms (metabolite, abbreviations) and equations (e.g. rates, processes) with the corresponding biochemical parameter would help to keep an overview of definitions, equations and context.

1)     Fig. 1. needs graphical improvements, e.g. by increasing the font size of the relevant enzymes. The letters m and n probably mean the number of substrate molecules bound to the enzyme, not the number of subunits that the enzyme complex is composed of. Perhaps the relevant metabolites mentioned in the text could be included with the parameters.

2)     In Table 1: which metabolites are considered for the Initial Concentration? What is the source or basis for the chosen metabolite levels or assumed rates? Concerning X and Y: do you mean the metabolites downstream of the enzymes?

3)     Table 2 needs revision: the metabolites and the justification for their chosen concentrations should be referenced (Lactate is mentioned only in the Discussion). Moreover, the concentration unit has the format [mM] (not e.g. G/mM, or K/mM, which indicates a ratio of two parameters). This format also pertains to the labeling of the figures. The biochemical processes denoted by the kinetic parameters K1-4 and the term a (Eq 1) need to be better explained. Indicate the metabolic conditions here instead of referring to the Fig. 4 and 5. (See ref. 58 as an example)

3.                    Results

Please begin each section/paragraph with a sentence introducing the thought/ aim / focus of the following text.

1)     In Line 166: provide the glucose concentrations measured in the medium. "Significant amount" is not an appropriate term here.

2)     Section 3.2: this work has been already published – it is not well justified why the same graphs (Fig. 3A-D) are presented here again. What is the new aspect?

3)     Section 3.3: Again, explain first which gain of knowledge is to be acquired for the glycolytic regulation, in particular the two enzymes, with the simulations.

4)     Line 232, Fig. 4: to what does "enzyme activity" refer? Does it involve both PFK and PK; how?

5)     Line 234: This sentence is biochemically an over-interpretation; remove it.

6)     Line 238f: The results would be better placed in a table

Have the authors measured the actual activities of PFK and PK of the cells in the different conditions? Do the averages concentrations of NADH and NAD+ change with the glucose conditions? Are they the same in the different cells types? Such biochemical assays would allow to draw more stringent conclusions and support the validity of the mathematical simulation, more so than citing the literature in general terms.

7)     Line 251: This sentence on "inhibitory feedback mechanism" is confusing. How does lactate come in?  What part of Fig. 4. is meant? This section needs a better biochemical translation.

8)     Line 264: what were the glucose concentrations measured?

9)     Fig. 5: as mentioned before, concentrations have the format [mM]

4.                    Discussion

The Discussion is not well-structured and needs substantial improvements.  Firstly, it should start with recapitulating the aim of this work and summarize the key findings, before going on to placing the results into the more general relevance and context of the scientific literature. Restate briefly the key features of the "present mathematical model".

1)   Oscillation of glycolysis has also been studied by Wenbo et al, which deserves to be related to the authors' results and views.

Ref: Wenbo, L.; Wang, J. Uncovering the Underlying Mechanisms of Cancer Metabolism through the Landscapes and Probability Flux Quantifications. iScience 2020, 23, 101002, doi:10.1016/j.isci.2020.101002

2)    Line 285f: This paragraph is not clear and needs fundamental revision. Why discuss the effects of cyanide, which is an inhibitor of OSPHOS? The relationship of glycolysis to mitochondrial metabolism is not clearly described and should be combined with the paragraph below (Line350). Also, it should be noted that changes in NADH levels do not result only from glycolysis. Moreover, extracellular lactate can enhance cellular NADH levels. This needs to be considered in the interpretations.

3)    Line 302f: This paragraph is confusing and also needs revision. The choice of intracellular lactate is critical for the proposed inhibition; check the literature for real values in cancer cell lines of different malignancy. See for example:

Aboagye, E.O.; Mori, N.; Bhujwalla, Z.M. Effect of malignant transformation on lactate levels of human mammary epithelial cells. Advances in Enzyme Regulation 2001, 41, 251-260.

Grashei, M.; Biechl, P.; Schilling, F.; Otto, A.M. Conversion of Hyperpolarized [1-(13)C]Pyruvate in Breast Cancer Cells Depends on Their Malignancy, Metabolic Program and Nutrient Microenvironment. Cancers (Basel) 2022, 14, doi:10.3390/cancers14071845.

4)       Line 316f: "The Warburg effect is not significant..." Remove this unqualified sentence.  (Avoid using the term Warburg effect when not specifically defined.)

5)       Line 352: This sentence makes no sense (..? "without entering the TCA cycle.." - check the grammar).

5.                    Material and Methods

1)     Line 372: mention the serum addition (source, %)

2)     Lien 374: express glucose-concentrations in mM

3)     Line 379: what is the cell density on the growth surface?

4)     Line 379: Why was DPBS used for the autofluorescence measurements? The buffer contains no other nutrients (glutamine, alanine etc.), which may alter glucose uptake and metabolic activity. Instead, DMEM (or a similar pH-controlled medium without phenol red) should be used. Why were measurements performed at pH 6.9?

5)     At which wavelengths were the optical measurements performed?

6)     Line 386: why were Hela cells cultivated at 1g/L (mM?) instead of the standard high glucose DMEM?

7)     How often were these experiments repeated?

6.                    Conclusions

Some conclusions are not scientifically stringent. In particular, the statement of Line 428f is an over-interpretation.

This manuscript, while being very interesting and addressing a new perspective on the glycolytic activity in cancer cells, needs substantial revision to convey the gain of knowledge in a sound and clear manner.

The English is very good. Some minor editing will be required.

Round 2

Reviewer 1 Report

The authors have adequately addressed my concerns.

One final check for minor English usage is recommended. 

Author Response

Dear the Reviewer 1,

The comments of the reviewer were highly insightful and enabled us to improve the quality of our manuscript.

Thank you.

Best regards,

Takashi Amemiya

Reviewer 2 Report

The manuscript has been substantially improved – being more clear and informative. The authors addressed all the critical issues and suggestions. Especially useful are the new Table 2 defining the different parameters for the modelling of oscillations, and the added Table 4 giving a numeric overview oscillation data.

Just some comments –

·        Paragraph starting line 61f could be condensed (with less details); what is the basic message?

·        Table 1 (and the rest of the text) would profit by listing here the definitions of terms in the rate equations (otherwise scattered in different parts of the text: G, X, Y; Jin,... n1,2,.. etc)

·        In Table 2 and the biochemical definitions:

o   The term "changed " probably means "variable"?

o   "affinity constant of ...glucose" – affinity to what?

o   What does the superscript in " 1.0 mMm denote?

·        In Figure 4,  "phase diagram spanned by the (which?) enzymatic activity a" – does this refer to the combined four reaction rates n1,2...?

·        Paragraph line 419f needs editing; the last sentence is cryptic. What is the message concerning lactate – referring back to line 403?

On the whole, the manuscript needs only some minor English editing.

[As a note: the Word function "track changes" (all visible in the online pdf-file) made it very confusing, very cumbersome and time-consuming to read through the manuscript; it was difficult to distinguish between original and revised tables and parts of the text. Also some figures appeared in duplicate and were misplaced.]

The English is quite good. Only minor changes are required when edited.
